# Thermal Behavior of Single-Crystal Diamonds Catalyzed by Titanium Alloy at Elevated Temperature

**Pengyu Hou [1], Ming Zhou [1,*] and Haijun Zhang [2]**

[1]  School of Mechanical and Electrical Engineering, Harbin Institute of Technology, Harbin 150001, China; 14b908068@hit.edu.cn

[2]  Research Center of Laser Fusion, China Academy of Engineering Physics, Mianyang 621900, China; luckynavyboy@163.com

\*  Correspondence: zhouming@hit.edu.cn; Tel.: +86-137-9660-5358

**Abstract:** Single-crystal diamonds are considered as the best tool material for ultra-precision machining. However, due to its low thermal conductivity, small elastic modulus and strong chemical activity, titanium alloy has poor machinability and is a typically difficult-to-machine material. Excessive tool wear prevents diamonds from cutting titanium alloy. This study conducts a series of thermal analytic experiments under conditions of different gas atmospheres in order to research the details of thermochemical wear of diamonds catalyzed by titanium alloy at elevated temperatures. Raman scattering analysis was performed to identify the transformation of the diamond crystal structure. The change in chemical composition of the work material was detected be means of energy dispersive X-ray analysis. X-ray photoelectron spectroscopy was used to confirm the resultant interfacial thermochemical reactions. The results of the study reveal the diffusion law of the single-crystal diamond under the action of titanium in the argon and air environment. From the experimental results, the product of the chemical reaction corresponding to the interface between the diamond and the titanium alloy sheet could be found. The research results provide a theoretical basis for elucidating the wear mechanism of diamond tools in the titanium alloy cutting process and for exploring the measures to suppress tool wear.

**Keywords:** diamond; thermochemical wear; diffusion; titanium alloy; chemical reaction; thermal analysis

## 1. Introduction

Single-crystal diamonds are often used in ultra-precision machining for components where an optical-grade surface finishes Are generally required. Titanium alloy has a series of advantages such as high specific strength, high temperature resistance and corrosion resistance. Hence, titanium alloys could be used to provide additional value in optical applications. Ti–6Al–4V is a commonly used titanium alloy that is very suitable for aircraft engines and airframe components. For example, integral blade rotors must be made of hard-to-cut alloys (Ti–6Al–4V and Inconel 718). It is difficult to machine due to elevated cutting forces, high temperature, limited cutting conditions and excessive tool wear [1]. Thin-wall structures are usually manufactured out of aluminum and titanium alloys and are widely used in the aeronautical sector. However, their machining presents serious challenges such as vibrations because of excessive tool wear and high cutting force [2]. The critical problem faced during the machining process of these superalloys is the concentration of heat in the cutting zone and the difficulty in dissipating it. The concentrated heat in the cutting zone has a negative influence on the tool life [3]. Polvorosa carried out an experimental to study the effect of lubricant pressure and material heat treatment on cutting forces and tool wear evolution on turning of super alloys [4]. Nabhani et al. conducted dry-cutting experiments of titanium alloy with cemented carbide, CBN and PCD tools

and measured the critical bonding temperatures of three tool materials and workpiece materials with quasi-static contact method to be 760 °C, 740 °C and 900 °C, respectively [5]. Paul et al. [6] classified titanium alloy as non-diamond turnable because of excessive tool wear. As Ti–6Al–4V has a low thermal conductivity (about 15 W/(m°C)), the cutting temperature on the interface is very high. High cutting temperature will exacerbate the reactivity between the titanium and diamond. This serves to accelerate the tool wear rate as stated by Ezugwu et al. [7]. Therefore, it is necessary to research the tool wear of Ti-alloy cutting with diamond tools.

A number of researchers have reported so far on wear characteristics of diamond tool in machining of Ti–6Al–4V.

Zareena et al. [8] studied the wear of single-crystal diamond tools in the ultra-precision cutting of titanium alloys. The results show that the cutting temperature and pressure on the contact surface between the diamond tool and the titanium alloy are the main causes of the diamond tool wear. The wear mechanism of the diamond tool is the graphitization of the diamond. N. Corduan et al. [9] studied the wear mechanism of a polycrystalline diamond (PCD) cutting tool when machining titanium alloy Ti–6Al–4V. He pointed out that the wear form of the PCD cutting tool is crater-wear of the rake face and uniform wear and groove wear of the flank face. The wear mechanism is adhesion, diffusion and oxidation under high temperature, and indicates that the groove on the PCD tool is caused by diamond phase-transition. Silva et al. [10] used PCD tool to carry out cutting experiments on titanium alloy Ti–6Al–4V. It was found that the main failure mode of PCD tool when cutting titanium alloy is flank wear. The wear mechanism of PCD tool is mainly adhesion and attrition. The study mentioned above suggest that the wear mechanism is not mechanical, but thermochemical one. Graphitization of diamond is the main form of diamond tool wear when cutting Ti alloy with a diamond tool.

Based on the above analysis, in this work, a series of thermochemical experiments were carried out to simulate the process of diamond wear during Ti alloy-cutting with diamond tools, in order to study the thermochemical wear mechanism of diamond at elevated temperature under the catalytic of titanium alloy. The findings of this research could provide a theoretical basis for reducing tool wear in cutting Ti alloy with a diamond tool.

## 2. Materials and Methods of Thermal Analysis Experiments

In this work, thermal analysis experiments were carried out using a TGA1600 simultaneous thermal analyzer with two functions, including differential thermal analysis (DTA) and thermal gravimetric (TG). The samples that consisted of diamond and Ti–6Al–4V were heated in a designated temperature range, as shown in Figure 1. The heating range was 293 K to 1473 K, and the heating rate was set to 10 K/min. Moreover, argon or air continuously penetrated into the crucible with a flow rate of 40 mL/min during the heating process. The diameter and thickness of Ti–6Al–4V sheet were 4 mm and 2 mm, respectively.

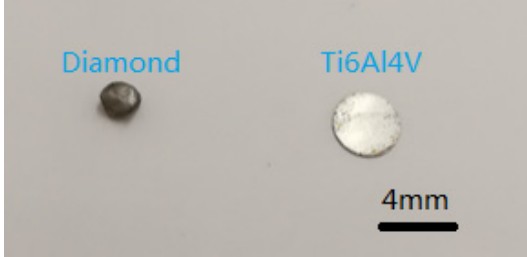

**Figure 1.** Diamond and Ti–6Al–4V workpiece used in thermal analysis experiment.

The reaction on the interface between Ti–6Al–4V and diamond samples was investigated by using a simultaneous thermal analyzer. Before each thermal analysis experiment, the diamond and titanium alloys were carefully cleaned in ultrasonic vessels with acetone. The diamond polished with

the designated crystallographic plane was set at the bottom surface of an aluminum oxide crucible in contact with the Ti–6Al–4V specimen.

## 3. Results and Discussion of Thermal Analysis Experiments

### 3.1. Diffusion of Carbon Atoms from Diamond into Titanium Alloy

Under high temperature, the carbon atoms in Ti alloy-cutting with the diamond tool easily diffuse into the titanium alloy workpiece on the tool/work contact interface in diamond-cutting of titanium alloys. Not only because there is a large difference in carbon concentration between titanium alloys and diamond crystals, but also because the atomic radius of the carbon atom is smaller than that of the titanium atom. This results in the diffusion wear of the diamond tool in cutting process.

In this work, the tool wear caused by the diffusion of carbon atoms from the diamond tool into the workpiece material during the titanium alloy cutting with diamond tool was studied. Scanning electron microscopy (SEM) and X-ray energy spectrum analyzer (EDS) were used to examine the energy spectrum of Ti–6Al–4V samples before and after the thermal analysis experiment. The micromorphology of Ti–6Al–4V samples in the contact zone with the diamond was also inspected after thermal analysis experiment.

Figure 2 shows the EDS diagram of the titanium alloy sample before and after the thermal analysis experiment in argon. It can be seen from Figure 2 that the carbon content on the surface of the titanium alloy sample significantly increased from 3.01% (before the experiment) to 23.31% (after the thermal analysis experiment), indicating that the diffusion reaction occurred at the contact interface between the diamond and titanium alloy flakes in the process of heating. It could be inferred that in the cutting process of titanium alloy by diamond tool, the carbon atoms in the diamond tool could diffuse into the titanium alloy material as long as the cutting temperature is high enough, which causes the diffusion wear of the diamond tool.

Figure 3 shows the microscopic morphology and EDS analysis data of the corrosion surface of titanium alloy sample in argon. It can be seen from Figure 3 that the carbon content of the white particles was as high as 28.2%, and it is inferred that the oxidation reaction between the carbon atom and titanium atom occurs, therefore, the surface of the titanium alloy flakes contained TiC after the reaction.

Figure 4 shows the EDS diagram of the titanium alloy sheet in the thermal analysis experiment in the air. It can be obtained from Figure 4 that the original carbon content of the titanium alloy sample before the test was 3.01%, and it increased to 4.52% on the corrosion surface after the experiment. This indicated that a slight diffusion activity of carbon atoms from the diamond crystal into the titanium alloy sample occurs during the heating process. Due to the presence of oxygen in the air, both the carbon atoms in the diamond and titanium atoms in the titanium alloy sample reacted with oxygen, which in turn, caused a certain influence on the diffusion reaction. In the air, the increase in the carbon content of the titanium alloy sample on the corrosion surface in the thermal analysis experiment was not large.

Figure 5 shows the microstructure and EDS analysis results of the corrosion surface of titanium alloy flakes in the air. It can be seen from Figure 5 that the white particles had an oxygen content of 28.2% and a carbon content of 5%, while the white particles in Figure 3 had an oxygen content of only 1.68%, from which it was inferred that the surface between titanium alloy flake and diamond occurred a serious oxidation reaction.

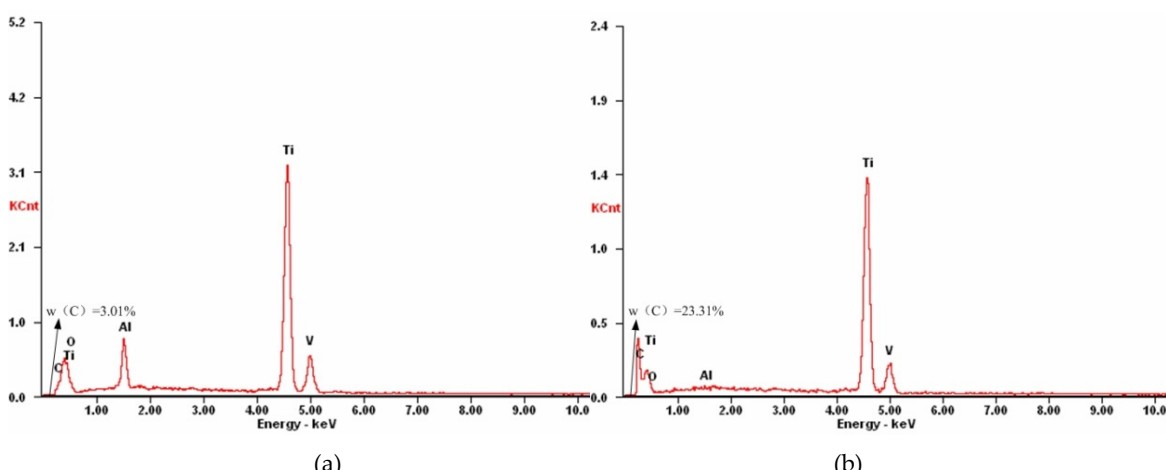

**Figure 2.** EDS analytical results of the titanium alloy sample surfaces before and after heated test in argon. (**a**) Before experiment; (**b**) after experiment.

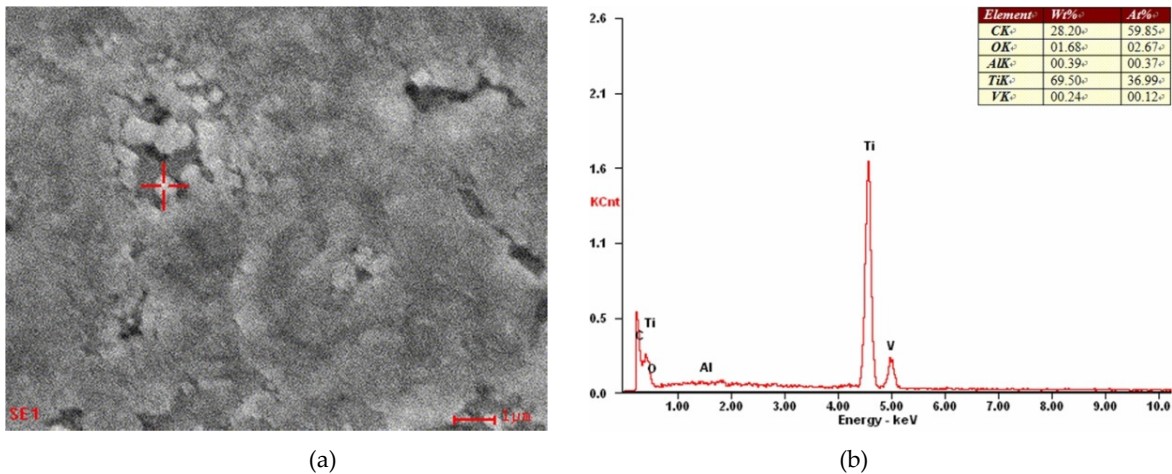

**Figure 3.** Micromorphology and EDS analysis of surface of titanium alloy after heated in argon. (**a**) Micromorphology; (**b**) EDS analysis.

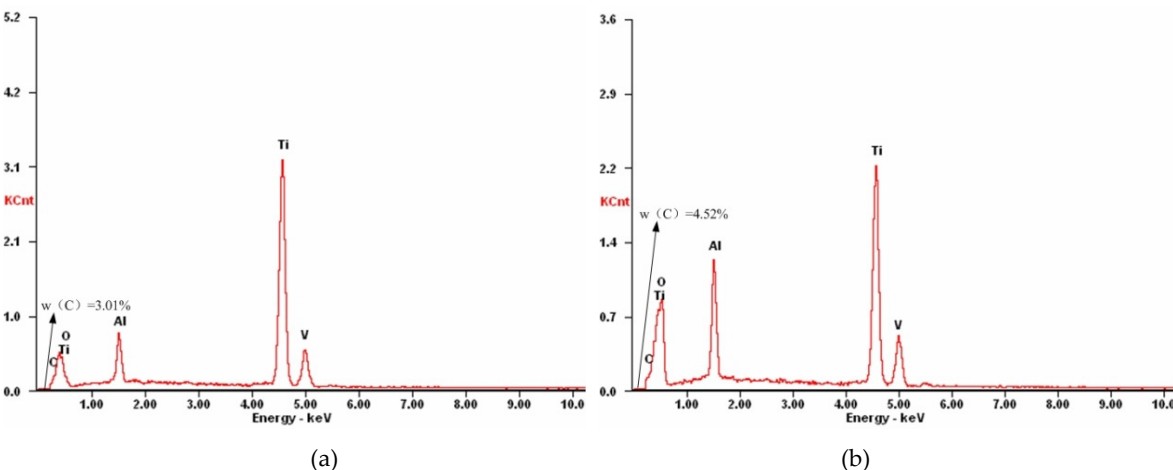

**Figure 4.** EDS of the titanium alloy sample surfaces before and after heated in air. (**a**) Before experiment; (**b**) after experiment.

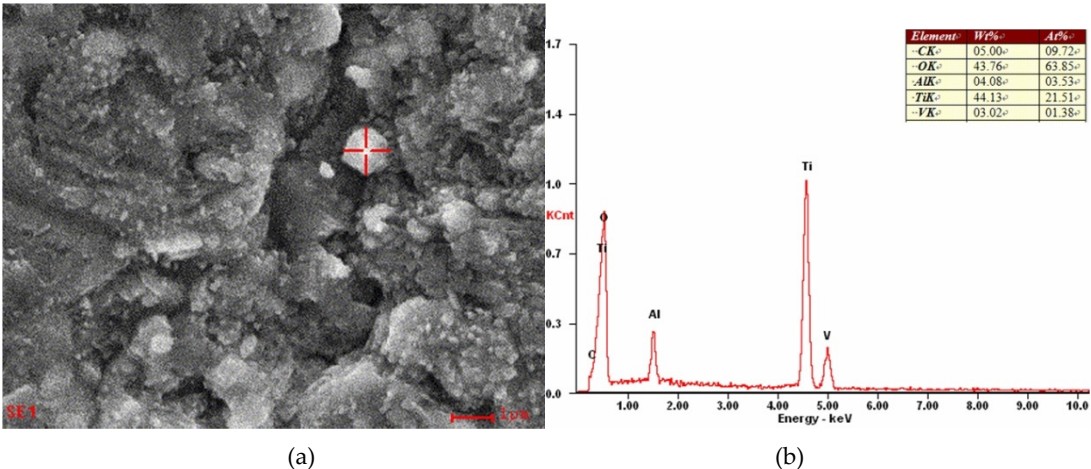

**Figure 5.** Micromorphology and EDS analysis of experimental surface of titanium alloy in air. (**a**) Micromorphology; (**b**) EDS analysis.

### 3.2. Thermodynamic Analysis of Diffusion Wear of Diamond Tools

Diffusion flux is the flow of diffused material through a unit cross-sectional area perpendicular to the direction of diffusion per unit time. The first law of diffusion states that the diffusion flux is proportional to the concentration at the cross section, and the larger the concentration gradient, the larger the diffusion flux. Hence, as long as there is a concentration difference, diffusion between atoms will occur, and the higher the temperature, the greater the diffusion rate. In the cutting process, the diamond tool is always in close contact with the fresh surface of the workpiece. Under the action of cutting heat and concentration gradient, the carbon atoms in the diamond tool will diffuse into the titanium alloy workpiece.

At the same time, during the cutting process, the continuous flow of the workpiece material in the cutting deformation zone keeps a large diffusion flux between the contact interfaces, and the strong plastic deformation of the workpiece material increases the dislocation density and the void. This further increases the diffusion of the tool material.

At a certain temperature, if the concentration of the diffusion element in the workpiece material exceeds its solubility, the diffusion element from the tool material will react with the workpiece material, and the diffusion reaction will be converted into a chemical reaction. Among the factors influencing the interdiffusion of tool and workpiece during the cutting process, temperature is the main factor, and its influence level is greater than the influence of the metal lattice structure. Therefore, Gibbs free energy can be used to judge the law of diffusion of tool materials [11]. Through the calculation, the correct change law can be obtained, and the diffusion wear of the diamond tool can be qualitatively analyzed [12].

The free energy of diamond tool material can be calculated as follows [13]:

$$\Delta G_f^0 = \Delta G^{XS} + RTlnc \qquad (1)$$

where $\Delta G^{XS}$ is the excess free energy of diamond tool materials in titanium alloy materials; $\Delta G_f^0$ is the free energy generation of diamond tool material; T is the temperature; R is the universal gas constant; c is the solubility of diamond tool material in workpiece material.

According to the reverse method [14], the excess free energy of carbon in the titanium alloy at 920 °C is 39.727 kJ/mol, which calculated by the Equation (2). The solubility of diamond tool materials in titanium alloy at different temperatures can be obtained by substituting this value into Equation (2) with the data in Table 1, as shown in Table 2.

According to Equation (1), c can be calculated as follows:

$$c = e^{\frac{\Delta G_f^0 - \Delta G^{XS}}{RT}} \tag{2}$$

In the Ti–6Al–4V, the mass of $\alpha$-titanium accounts for 80% of the total mass, so the solubility of the diamond tool material in $\alpha$-titanium is mainly considered. Carbon has a solubility of 1% in $\alpha$-titanium at 920 °C [15] Diamond is a metastable lattice structure of carbon, and graphite is a stable lattice structure of carbon.

Under certain conditions, the crystal lattice of carbon atoms in diamonds is automatically converted into a graphite structure, that is, graphitization of diamond.

At the same time, since the carbon atoms in the diamond are arranged in a denser arrangement than the carbon atoms in the graphite structure, the diffusion of carbon atoms is bound to occur after the graphitization of the diamond [6].

The Gibbs generation free energy of carbon in diamond tools at different temperatures is calculated by the first approximate equation of Gibbs–Helmholtz as Equation (3) [16], that is, the Gibbs generation free energy of diamond into graphite.

$$\Delta G_T^{\ominus} = \Delta H_{298}^{\ominus} - T\Delta S_{298}^{\ominus} \tag{3}$$

where $\Delta G_T^{\ominus}$ is the standard Gibbs free energy at temperature $T$; $\Delta H_{298}^{\ominus}$ is the standard molar enthalpy of formation of the reaction substance at room temperature; $\Delta S_{298}^{\ominus}$ is the Gibbs standard reaction entropy difference at room temperature.

As described earlier, the excess free energy of diamond tool materials in titanium alloy materials at different temperature are calculted as shown in Table 1.

**Table 1.** Gibbs free energy of carbon under different temperatures.

| T/K | 298 | 500 | 700 | 900 | 1100 | 1300 | 1500 |
|-----|-----|-----|-----|-----|------|------|------|
| $\Delta G_f^0$ | −2.8929 | −3.5775 | −4.2553 | −4.9331 | −5.6109 | −6.2887 | −6.9665 |

Table 2 shows that as the temperature increases, the solubility of carbon in the titanium alloy increases, which is also consistent with the first law of diffusion, indicating that it is feasible to use the Gibbs free energy to qualitatively analyze the diffusion process. Hence, the higher the cutting temperature, the more severe the diffusion wear of diamond tools in ultra-precision cutting of titanium.

Therefore, in order to improve the life of the diamond tool, the cutting temperature must be reduced.

**Table 2.** Solubility of carbon in titanium alloy under different temperatures.

| T/K | 298 | 500 | 700 | 900 | 1100 | 1300 | 1500 |
|-----|-----|-----|-----|-----|------|------|------|
| c | $3.3817 \times 10^{-8}$ | $2.9912 \times 10^{-5}$ | $5.2225 \times 10^{-4}$ | $2.6 \times 10^{-3}$ | $7 \times 10^{-3}$ | $1.42 \times 10^{-2}$ | $2.37 \times 10^{-2}$ |

### 3.3. Study on Chemical Reaction of Titanium Alloy and Diamond in Air and Argon

By determining the binding energy of carbon atoms and titanium atoms in different valence states, the composition of carbon and titanium compounds is determined, and the type of chemical reaction between the titanium alloy and the diamond interface is confirmed.

Figure 6 shows an electron spectrum of the characteristic peak of Ti2p obtained by XPS analysis of the reaction surface between the titanium alloy sheet and diamond particles in argon after hot corrosion reaction. According to the original line of Ti2p in Figure 5, spin splitting of Ti2p level orbital produces two peaks of Ti2p1/2 and Ti2p3/2. Therefore, each compound of Ti or Ti elemental should also have

two peaks. The coincidence of the original line of Ti2p and the fitted line in Figure 6 shows that the spectrum obtained by peaking the peak value of XPS is reasonable, and the combined energy of the eight peaks is 453.7 eV, 454.4 eV, 455.2 eV, 457.3 eV, 459.4 eV, 460.1 eV, 460.9 eV and 463 eV, respectively. Referring to the relevant data, it can be inferred that they correspond to Ti, TiC, $Ti_2O_3$, $TiO_2$, Ti, TiC, $Ti_2O_3$ and $TiO_2$, respectively. This also proves that carbon atoms react with titanium atoms to form TiC, which causes chemical wear of diamond.

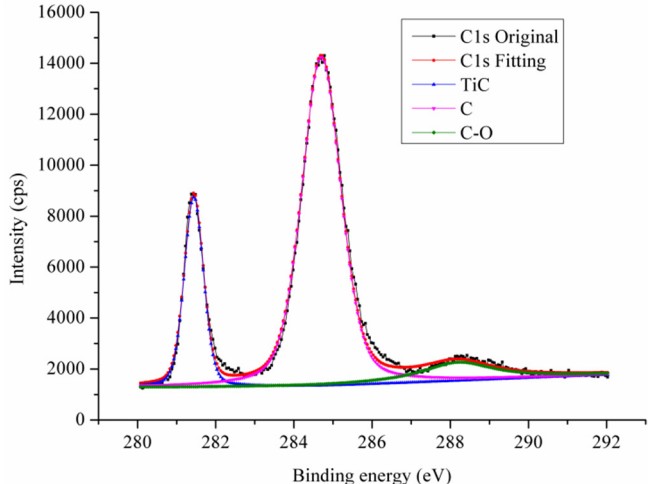

**Figure 6.** Electronic spectra of C1s characteristic peak on corrosion interface of titanium alloy in argon.

Since the intensity of the peak corresponds to the content of the reaction product on the corrosion interface of the titanium alloy, it can be inferred from Figures 6 and 7 that the content of the material on the hot corrosion interface of the titanium alloy sheet is Ti, $TiO_2$, C, TiC and $Ti_2O_3$ from high to low. It shows that in the anaerobic environment, due to the catalytic action of titanium, the diamond particles undergo a graphitization transformation. A small portion of the generated graphite atoms react with titanium to form TiC, and the remaining carbon atoms either diffuse into the titanium alloy matrix or accumulate at the reaction interface to form a graphite layer.

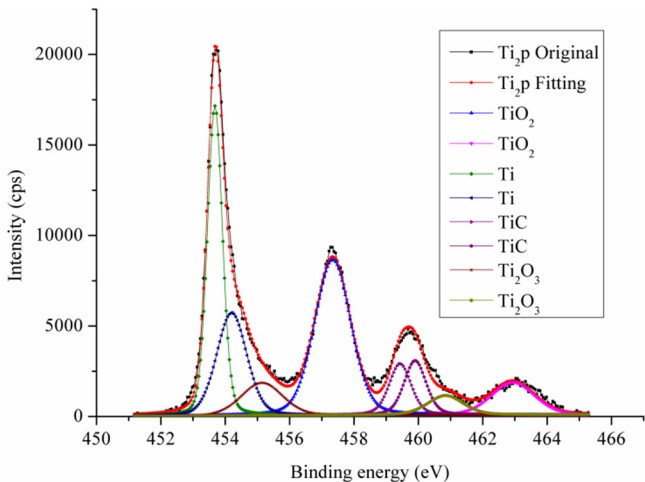

**Figure 7.** Electronic spectra of Ti2p characteristic peak on corrosion interface of titanium alloy in argon.

Therefore, the diamond tool has diffusion wear and oxidative wear, and it mainly focuses on diffusion wear. Due to the strong chemical activity of Ti, before the hot corrosion experiment, a thin film of Ti was formed on the titanium alloy sheet, which contains $TiO_2$ and $Ti_2O_3$. Since the binding ability of titanium with oxygen is stronger than that of titanium and carbon, the oxide of titanium is basically unchanged during the thermal analysis experiment in argon.

Figure 8 is an electron spectrum of the characteristic peak of O1s obtained by XPS analysis of the reaction surface of the titanium alloy sheet after hot corrosion reaction with diamond particles in air. It can be inferred from the coincidence degree of the original line of O1s and the fitted line in Figure 8 that the two lines obtained by peaking the peak value of the XPS spectrum are reasonable, and the binding energy of the two peaks correspond to 529.6 eV and 531.4 eV, respectively. Which correspond to the $TiO_2$ and C–O bond lines, respectively.

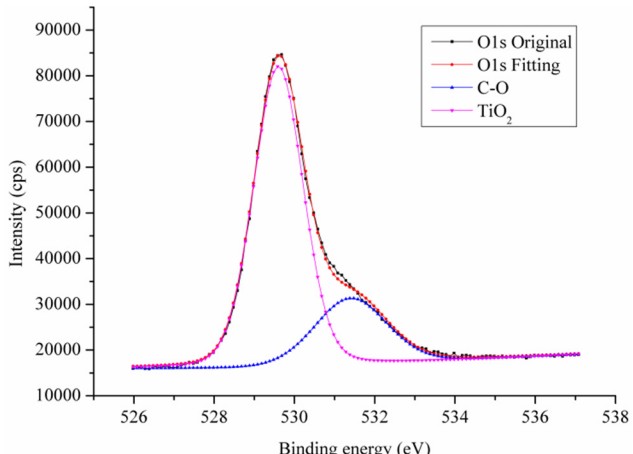

**Figure 8.** Electronic spectra of O1s characteristic peak on corrosion interface of titanium alloy in air.

Figure 9 is an electronic spectrum of the characteristic peak of O1s obtained by XPS analysis. It is known from the original line of Ti2p that spin splitting of the 2p level orbital of Ti produces two peaks of Ti2p1/2 and Ti2p3/2. From the coincidence degree of the original line of Ti2p and the fitted line in Figure 9, it is known that the peak obtained by peaking the peak value of XPS spectrum is reasonable, and the combined energy of the two peaks is 457.3 eV and 463 eV, both corresponding to $TiO_2$. The intensity of the peak corresponds to the content of the reaction product on the corrosion interface of the titanium alloy.

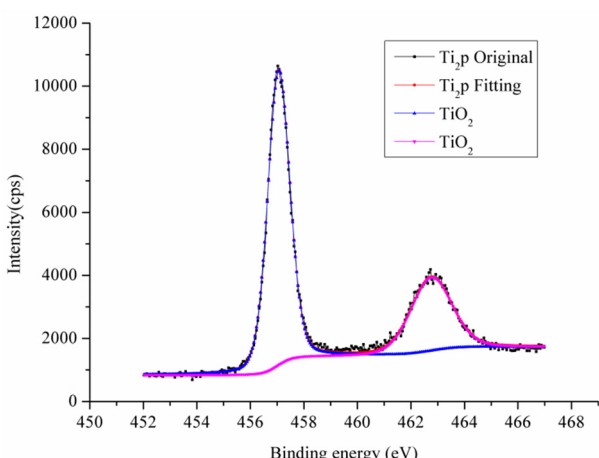

**Figure 9.** Electronic spectra of Ti2p characteristic peak on corrosion interface of titanium alloy in air.

It can be inferred from Figures 8 and 9 that the hot corrosion interface of titanium alloy flakes is almost entirely $TiO_2$, which proves that in the aerobic environment, the oxidation wear of diamond tools is formed by the chemical reaction of carbon atoms and oxygen atoms to generate CO and $CO_2$.

## 4. Conclusions

In the process of Ti alloy-cutting with diamond tool, the wear of the diamond tool can be summarized as follows:

(1) In absence of oxygen at high temperatures of the cutting zone and the catalysis of titanium atoms, the carbon atoms in the single-crystal diamond tool escape from the crystal lattice of the diamond and transform into a graphite structure. Most of the escaping carbon atoms diffuse into the titanium alloy workpiece material, contributing to the diffusion wear of the diamond tool. The higher the temperature, the more severe the diffusion wear of the diamond tool, and a small part of the oxidation reaction with titanium generates TiC, which forms the oxidative wear of the diamond tool;

(2) When oxygen is present at relatively low temperatures, carbon atoms that are shed from the diamond lattice can react with oxygen to form CO and $CO_2$, forming oxidative wear of the diamond tool. The higher the temperature, the more severe oxidative wear of the diamond tool. Therefore, reducing the temperature in the cutting zone of the tool with the titanium alloy workpiece material is a fundamental measure to suppress the wear of the diamond tool.

**Author Contributions:** P.H. and M.Z. conceived and designed the experiments; P.H. performed the experiments; P.H. and H.Z. analyzed the data; M.Z. contributed the materials; P.H. wrote the study. All authors have read and agreed to the published version of the manuscript.

**Funding:** This work was supported by the research fund from the National Natural Science Foundation of China (Grant No. 51975153).

**Conflicts of Interest:** The authors declare no conflict of interest.

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
