# Peer review of "Thermal Behavior of Single-Crystal Diamonds Catalyzed by Titanium Alloy at Elevated Temperature"

_applsci, doi:10.3390/app10134651_

Round 1
Reviewer 1 Report
Nice little work. However, need some major English/grammatical corrections. The use of preposition is not correct most of the time. Below I have included some but there are many more corrections needed to be done throughout the manuscript.
1. Line 21-23: The long sentence needs re-structuring and simplification.
2. Line 36-37: The sentence should start with "As".
3. In general throughout the manuscript I recommend replacing the word "cutter" with "cutting tool" (example line 46, 47)
4. In general throughout the manuscript I recommend replacing "Diamond cutting of Ti alloy" with "Ti alloy cutting with diamond tool" (example line 39, 56, 59)
5. The term "hot corrosion test" appeared suddenly. It is not clear which test you are referring to as a "hot corrosion test".
6. Line 76-79: long sentence, please re-structure.
7. Line 81-82: grammatical mistakes, please modify
8. Figure 2,4: Are these just single point analysis or did you do surface mapping? If just single point analysis then I would like to see more points (may be in a table form will do), or EDS mapping of the surface which will show how the %C has increased throughout the surface.
9. Figure 3, 5 basically based on single point analysis. Drawing conclusion from just a single point is not enough.
10. Line 188, section heading 3.3: you have just used Ar and Air. So, just mention those instead of using "different gas"
11. I am not comfortable in using terms like "anaerobic and aerobic reactions" mainly because these are more like biological terms. May be using "in presence/absence of Oxygen" (optional)
Reviewer 2 Report
The contribution is applicable for journal after revision.
Expand the titles of chapters.
Please, correct many typographical errors and typos.
Correct units, indexes and so on.
The authors must revise the manuscript. I enclosed the paper with notes.

Author Response
1.The titles of chapters has been expand: "of Thermal Analysis Experiments" has been added in line 62, 76.
2.According to the Suggestions of reviewers, Spaces have been added in line 35, 37, 40, 43, 51, 66, 67, 68, 69, 92, 100, 105, 114, 115, 155, 165, 171, 173, 179, 186, 226.
3.Indexes has been corrected in line 195, 197, 198, 199, 203, 209, 210, 217, 227, 231, 232, 235, 240, 254.
4.The word "Origiral" has been replaced with "Original" in Figure 6, 7, 8, 9.
5.A ruler has been added in Figure 1。